# 100-Day Mission for Future Pandemic Vaccines, Viewed Through the Lens of Low- and Middle-Income Countries (LMICs)

**DOI:** 10.3390/vaccines13070773

**Published:** 2025-07-21

**Authors:** Yodira Guadalupe Hernandez-Ruiz, Erika Zoe Lopatynsky-Reyes, Rolando Ulloa-Gutierrez, María L. Avila-Agüero, Alfonso J. Rodriguez-Morales, Jessabelle E. Basa, Frederic W. Nikiema, Enrique Chacon-Cruz

**Affiliations:** 1Escuela de Medicina, Universidad de Monterrey, San Pedro Garza García, Nuevo León 66238, Mexico; yodira.hernandez@udem.edu; 2Think Vaccines LLC., Houston, TX 77005, USA; zlopatynsky@gmail.com; 3Servicio de Aislamiento Pediatría, Hospital Nacional de Niños “Dr. Carlos Sáenz Herrera”, San José 1654-1000, Costa Rica; rolandoug@gmail.com; 4Cátedra de Pediatría, Universidad de Ciencias Médicas (UCIMED), San José 10108, Costa Rica; 5Servicio de Infectología, Hospital Nacional de Niños “Dr. Carlos Sáenz Herrera”, San José 10905, Costa Rica; avilaaguero@gmail.com; 6Universidad de Costa Rica, San José 10905, Costa Rica; 7Faculty of Health Sciences, Universidad Científica del Sur, Lima 15307, Peru; arodriguezmo@cientifica.edu.pe; 8Grupo de Investigación Biomedicina, Faculty of Medicine, Fundación Universitaria Autónoma de las Américas-Institución Universitaria Visión de las Américas, Pereira 660003, Colombia; 9Research Institute for Tropical Medicine, Muntinlupa City 2428, Metro Manila, Philippines; jessabelle.basa@ritm.gov.ph; 10Institut de Recherche en Sciences de la Santé, Direction Régionale de l’Ouest, 399 Avenue de la Liberté, Bobo-Dioulasso 01, Hauts-Bassins 0P1 BP 545, Burkina Faso; dr.nikiema@gmail.com

**Keywords:** vaccines, pandemic, 100-day plan, CEPI

## Abstract

The 100-Day Mission, coordinated by the Coalition for Epidemic Preparedness Innovations (CEPI) and endorsed by significant international stakeholders, aims to shorten the timeframe for developing and implementing vaccines to 100 days after the report of a new pathogen. This ambitious goal is outlined as an essential first step in improving pandemic preparedness worldwide. This review highlights the mission’s implementation potential and challenges by examining it through the lens of low- and middle-income countries (LMICs), which often face barriers to equitable vaccine access. This article explores the scientific, economic, political, and social aspects that could influence the mission’s success, relying on lessons learned from previous pandemics, such as the Spanish flu, H1N1, and COVID-19. We also examined important cornerstones like prototype vaccine libraries, accelerated clinical trial preparedness, early biomarkers identification, scalable manufacturing capabilities, and rapid pathogen characterization. The review also explores the World Health Organization (WHO) Pandemic Agreement and the significance of Phase 4 surveillance in ensuring vaccine safety. We additionally evaluate societal issues that disproportionately impact LMICs, like vaccine reluctance, health literacy gaps, and digital access limitations. Without intentional attempts to incorporate under-resourced regions into global preparedness frameworks, we argue that the 100-Day Mission carries the risk of exacerbating already-existing disparities. Ultimately, our analysis emphasizes that success will not only rely on a scientific innovation but also on sustained international collaboration, transparent governance, and equitable funding that prioritizes inclusion from the beginning.

## 1. Historical Background

The Spanish flu of 1918 was one of the deadliest pandemics in history, claiming about 50 million lives worldwide. This event remains a cornerstone for understanding pandemics, especially since it exposed the devastating impact of inadequate public health infrastructure. Poor housing, overcrowding, military conflicts (e.g., the First World War and the Mexican Revolution), and a lack of resources significantly contributed to the pandemic’s severity [1]. At the same time, public health initiatives such as enhancing hygiene and promoting social distancing emerged as efforts to control the outbreak. However, vaccination research was still in its early stages, as immunology and virology would only develop much later, making it possible to create targeted vaccinations [2].

Fast forward to the spring of 2009, when the World Health Organization (WHO) announced the AH1N1 influenza pandemic, it served as yet another reminder of these vulnerabilities. This time, researchers developed a vaccine, but it proved insufficient to halt the early stages of the pandemic [3]. Millions became infected due to the delayed production and distribution of vaccines, healthcare systems were strained, and economic losses escalated as infection and fear rates increased [3]. The Centers for Disease Control and Prevention (CDC) and the WHO estimate that between 151,700 and 575,400 people died worldwide during the first year of the AH1N1 pandemic [3]. Khazeni et al. estimated that if 20% of Americans had received vaccines in October, 1067 deaths may have been prevented, and USD 159,000,000 could have been saved, compared to no vaccination. These figures demonstrate that even a slight increase in vaccine distribution can have a substantial impact on the economy, the transmission rate, and the number of lives saved [4]. Although the global fatality rate was lower than previously anticipated, it emphasized a harsh reality: without faster vaccine production, even relatively mild pandemics can have outsized social and economic consequences [5].

Severe Acute Respiratory Syndrome (SARS) emerged as a novel human disease in late 2002. The first known case occurred in southern China in November 2002, and the virus was brought to Hong Kong on 21 February 2003 by an infected patient. This led to ten secondary cases, triggering outbreaks in two hospitals in Hong Kong and further spread to Singapore, Toronto, and Hanoi. By March 2003, a novel coronavirus (SARS-CoV) was identified as the causative agent. Within 11 weeks of the first case in Hong Kong, SARS had spread to 27 additional countries or special administrative regions. The mini-pandemic peaked in the last week of May 2003, with the final probable case reported on 13 July 2003. In total, there were 8096 probable cases and 774 deaths. Of these, 66% occurred in China, 22% in Hong Kong, 4% in Taiwan, and 3% each in Singapore and Canada. Notably, 21% of cases were among healthcare workers. Unlike COVID-19, which predominantly affected the elderly, SARS primarily impacted children [6].

Another significant event occurred in January 2020, when the WHO declared the COVID-19 pandemic a Public Health Emergency of International Concern. In contrast with previous pandemics, the scientific community achieved vaccine development through modern technology, such as mRNA platforms. COVID-19 vaccines were authorized for emergency use less than a year after the virus’s genetic sequence was studied, demonstrating the power of technology, innovation, and international cooperation [7]. Despite this, unequal vaccine distribution revealed political and economic divides, exposing low- and middle-income countries (LMICs) and highlighting the need for a more efficient and fair response system [7]. At the same time, the rapid development and deployment of vaccines during the COVID-19 pandemic underscored the critical need for vigilant post-marketing surveillance, also known as Phase 4 studies, to safeguard long-term safety and efficacy, particularly across varied demographic populations [8].

As the most efficient way to contain and manage infectious diseases, vaccines are the mainstay of contemporary defenses against pandemics. Time is of the essence: the quicker a vaccine is created and implemented, the quicker a possible pandemic may be contained.

The Spanish flu, AH1N1 influenza, and COVID-19 pandemics demonstrate that prompt and efficient reactions to new health threats are vital. Every incident highlighted different challenges, such as the poor public health infrastructure in 1918, the delayed vaccine rollout in 2009, and the uneven distribution of COVID-19 vaccines in 2021. All these experiences suggest that time and prompt preventive measures are one of the most critical components in pandemic control.

Based on these insights, the Coalition for Epidemic Preparedness Innovations (CEPI) disclosed the ambitious 100-Day Mission. This mission aims to transform global preparedness by creating vaccines within 100 days of discovering a new pathogen. It seeks to control the pandemic by utilizing scientific advancements and international cooperation before it worsens [9]. The following section examines the vision, structure, and potential impact of this revolutionary mission.

Nonetheless, there is currently no comprehensive overview of the 100-Day Mission from the perspective of LMICs, which account for nearly half of the global population when compared to high- and upper-middle-income countries [10].

## 2. The Overall Vision of the 100-Day Mission

The 100-Day Mission, led by CEPI and supported by international organizations such as the G7 and G20, is a bold attempt to transform vaccine development timeframes [9]. This project uses cutting-edge technology and global collaboration to reduce the risk of future pandemics by aiming to develop a vaccine for the next Disease X in as little as 100 days. This mission is to prevent outbreaks before they worsen, reducing their devastating impact on health, society, and the economy.

Motivated by lessons learned from COVID-19, the mission targets expanding global concerns in a constantly changing biosecurity environment, such as antibiotic resistance, bioterrorism, and emerging infectious illnesses [9,11,12]. Its vision is centered on expanding global participation, especially from LMICs, by strengthening regulatory coherence and establishing fair access frameworks.

The following are the key pillars to achieve the 100-Day Mission.

### 2.1. Pre-Existing Prototype Vaccines

The cornerstone of the 100-Day Mission is the foundation of a “vaccine library”, a collection of prototype vaccines prepared for quick adaptation and distribution against new threats. Li et al. investigate developments in next-generation vaccine technologies, such as mRNA platforms, to address future virus evolution and facilitate quick vaccine creation [13]. Verma et al. and Pogostin and McHugh emphasized the need for innovative vaccination adjuvants to increase immune responses and efficacy [14,15]. Although mRNA technology is essential for rapid manufacturing, the goal incorporates several platforms to guarantee adaptability [16]. These technical advancements, rapid diagnostic techniques, and genomic surveillance form the basis of the 100-Day Mission.

The vaccine library currently seeks to keep up with varying pathogens by focusing on viral families with demonstrated pandemic potential, such as *Paramyxoviridae*, *Flaviviridae*, *Togaviridae*, *Filoviridae*, *Bunyaviridae*, *Arenaviridae*, *Picornaviridae*, among others [17,18]. Its content will change to reflect newly discovered risks as scientific knowledge advances. Two significant obstacles to overcome in creating these prototypes are developing effective regulatory channels and scalable production techniques for rapid rollout.

### 2.2. Global Clinical Trial Infrastructure and Readiness

CEPI emphasizes that a global clinical trials network with pre-approved protocols is crucial for accelerating vaccine development. CEPI intends to facilitate rapid responses to new threats by integrating worldwide manufacturing facilities and accelerating clinical trial procedures. A robust clinical laboratory network ensures timely data readouts, enabling prompt assessments of vaccine efficacy and safety [19].

Additionally, CEPI is creating a collection of prototype vaccines targeting important virus families, tested through Phase 1 clinical trials, to be able to adapt and respond to emerging threats [20]. This initiative enhances the global ability to adapt swiftly to new threats while maintaining a high standard of safety and efficacy.

The effectiveness of the library relies on ongoing collaboration between international organizations, public health agencies, and scholars. This procedure entails gathering and sequencing viral samples, monitoring emerging infectious diseases in real time, and tracking genetic diversity [21]. To manage enormous amounts of genomic data, spot possible pandemic risks, and direct vaccine research efforts, sophisticated bioinformatics tools and data analytics are essential [12]. The genomic and vaccine libraries are dynamic, continuously updated resources. By staying current with the most recent information, the international community guarantees that they are prepared to react rapidly and effectively to pandemics in the future, protecting public health and causing the least disturbance to society [21]. To date, vaccine libraries within the context of the 100-Day mission remain largely absent in LMICs.

### 2.3. Earlier Biomarkers of Immune Response

Accelerating vaccine timelines requires the identification of early biomarkers of a strong immune response and protection. Instead of the conventional 14–21-day period, immunological markers can now offer quicker signs of vaccine effectiveness. This invention boosts trust in vaccination deployment during crucial times while reducing delays [19], a gap that remains inconsistently addressed across LMICs.

### 2.4. Rapid Manufacturing Capacity

CEPI aims to streamline manufacturing procedures for quick initial and scale-up output. Within days, rapid activation is guaranteed through a network of “warm” manufacturing facilities with capacity reserved for multiple platforms, which includes both the effort of high-income countries (HICs) and LMICs. Further bolstering rapid vaccine manufacture and delivery are developments in next-generation vaccine technologies, including mRNA [19].

### 2.5. Early Characterization of Pathogens

Enhancements in surveillance capacity are crucial for early detection and response to outbreaks. CEPI prioritizes standardizing global procedures for sharing genetic sequences and initiating outbreak alarms. During the early stages of an outbreak, sophisticated technologies like in silico modeling are essential for determining correlates of protection and evaluating possible toxicity [19]. These innovative techniques offer crucial information to support prompt and efficient reactions.

This strategy focuses on proactively investigating and creating medicinal countermeasures for virus families with high pandemic potential, such as vaccinations and monoclonal antibodies, before an outbreak [18]. Viral behavior, pathogenesis, and possible targets for intervention are better understood by researchers when they concentrate on “prototype pathogens” that are representative of each virus family [22]. When a new virus appears in that family, this existing information, if performed uniformly in both HICs and LMICs, will dramatically accelerate the development process.

## 3. Benefits Across Sectors

### 3.1. Economic Impacts of the 100-Day Mission

Besides being a revolutionary method for pandemic preparedness, the 100-Day Mission is also a crucial economic strategy. Pandemics have historically caused significant financial damage. For example, the COVID-19 pandemic is estimated to have cost the global economy USD 12.5 trillion by 2024, due to lost productivity, healthcare expenses, supply chain disruptions, and decreased customer confidence [23]. Additionally, it resulted in a 3.4% decline in global gross domestic product (GDP) in 2020, equivalent to over USD 2 trillion in lost output [24]. Had countries implemented non-pharmaceutical interventions (NPIs) as effectively as in past pandemics, the 100-Day Mission could have averted an estimated 8.33 million deaths globally (95% credible interval [CrI]: 7.70–8.68 million), the majority of which would have occurred in LMICs. This impact translates into an approximate monetary benefit of USD 14.35 trillion (95% CrI: 12.96–17.87 trillion), based on the value of statistical life-years saved. Moreover, additional investments in manufacturing capacity and health systems could have increased the number of lives saved to 11.01 million (95% CrI: 10.60–11.49 million) [25].

Immediate vaccination rollouts can save lives and contribute to economic recovery. Significant differences in the efficacy of vaccination distribution can be observed when examining past pandemics such as AH1N1. A postponed vaccination response during AH1N1 resulted in a heavy burden on healthcare systems and lost chances to stop deaths. On the other hand, the COVID-19 pandemic has brought to light the potential of rapid vaccine rollout to save billions of dollars in economic expenses and millions of lives.

Figure 1 compares the estimated outcomes of vaccine deployment during the 100-Day Mission, highlighting both lives saved and financial gains in U.S. dollars. As illustrated, the number of averted deaths would be greater in LMICs than in HICs. However, as expected—due to the higher value placed on statistical life in HICs—the projected financial gains from prompt vaccination are greater in those settings.

A key component of this program is its emphasis on advanced manufacturing technologies, such as adenoviral vector platforms, which enable fast and affordable vaccine production [16]. These developments increase the return on investments in pandemic preparedness initiatives and align with the mission’s timeframe.

International collaborations are essential to guaranteeing the equitable distribution of the resources and expertise required to implement the 100-Day Mission. International financial responsibilities, including investment in infrastructure, research, and innovation, address the difference between high- and low-income countries. During pandemics, these collaborations reduce inequities, guaranteeing universal vaccine access and promoting international economic stability, both for HICs and LMICs [21].

The strategy not only prevents economic crises but also builds the long-term resilience of global health systems, paving the way for sustainable growth. According to Farlow et al., preventing the unequal effects of pandemics on low-income countries requires fair access to vaccines [26]. According to Moore et al., removing these obstacles is crucial to ensuring the mission’s success and creating an inclusive economy that can withstand future medical crises [21].

The economic benefits of the 100-Day Mission in HICs extend across numerous sectors:Healthcare: Early vaccine deployment reduces hospitalizations and intensive care needs, resulting in billions of dollars in healthcare cost savings. For example, the U.S. government allocated USD 178 billion in funding to hospitals during COVID-19 [27]. Rapid containment through vaccination could substantially lower such expenditures.Tourism and Hospitality: Compared to the academic year 2019, the revenue of the worldwide travel and tourism industry declined by 34.7% in the first quarter of 2020. This quarter’s expected revenue of USD 447.4 billion was significantly lower than anticipated due to border closures and travel restrictions [28]. Early vaccines could mitigate these losses, preserving jobs and revenues.Education: Long-term school closures affected 1.53 billion students in 184 countries, a sizable percentage of students worldwide, resulting in long-term economic consequences due to reduced human capital [28]. Accelerated vaccine deployment could prevent such disruptions and safeguard future economic productivity.

In summary, the 100-Day Mission promises significant economic benefits, including reduced healthcare costs, sustained vital industries, and a more inclusive and resilient global economy. However, these precise analyses have not yet been estimated in LMICs.

### 3.2. Political Impacts of the 100-Day Mission

For the 100-Day Mission to succeed, unprecedented international collaboration is necessary, as adequate pandemic preparation requires the sharing of data, resources, and knowledge. A coordinated plan with worldwide participation is essential. Numerous studies have shown that success depends on political will, significant investments in public health infrastructure, and the guarantee of fair access to vaccines [9,20,29]. Additionally, governments must prioritize vulnerable people, overcome vaccine nationalism, and fund early warning systems.

Strong legislative frameworks that guarantee quick vaccine deployment, such as pre-negotiated contracts with pharmaceutical companies, expedited regulatory procedures, and investments in manufacturing infrastructure, are essential to the mission’s success. For international cooperation and the fair allocation of resources, geopolitical tensions and intellectual property disputes must be resolved.

Governments, pharmaceutical companies, and other stakeholders must openly manage conflicts of interest, as they can undermine public confidence and obstruct unbiased, evidence-based decision-making. Additionally, global collaboration and equitable resource distribution require resolving geopolitical difficulties and intellectual property barriers [30].

### 3.3. Social and Health Impacts of the 100-Day Mission

The 100-Day Mission prioritizes vaccine development and delivery within a strict timeframe, demonstrating a proactive commitment to preventing future pandemics. Its main objective is to guarantee that vaccines are prepared within 100 days of discovering a novel virus that can potentially spread like a pandemic [31]. This strategy aims to drastically shorten the interval between disease identification and vaccine availability, potentially saving millions of lives and averting the breakdown of healthcare systems during outbreaks.

Some significant concerns are addressing cold chain requirements, removing logistical obstacles in vaccine delivery, and guaranteeing fair vaccine access for LMICs [7,20,21]. To overcome these obstacles, initiatives such as international collaborations and the development of adaptable storage solutions are essential.

In HICs, by reducing the impact of pandemics on social structures, the workforce, and education, the strategy seeks to minimize social and economic upheavals in addition to health issues [9]. Accordingly, global cooperation, technological advancement, and dedication to resolving disparities in pandemic responses are essential to its success.

Socially and economically vulnerable individuals are most severely impacted by pandemics, which accentuate pre-existing disparities in gender, housing, income, and access to basic services. As Cuevas Barron et al. highlight, these groups are at higher risk of experiencing systemic exclusion, particularly if the health and social protection systems are fragmented or underfunded [32]. The 100-Day Mission must address these systemic biases by creating vaccination strategies that are both socially and medically inclusive. Achieving global vaccine equity, according to Figueroa et al., involves deliberate attempts to prioritize underserved regions, especially LMICs, and to address the obstacles that sustain exclusion [7]. Such strategies must include vulnerable groups, such as the elderly, migrants, informal workers, and those who face digital challenges, as emphasized by Cuevas Barron et al. [32]. In the absence of intentional promotion, pandemic responses risk repeating patterns of exclusion. Additionally, a respectful alliance and genuine community involvement are crucial in addressing social concerns such as vaccine hesitancy, mistrust, and inadequate health literacy. While concerns about vaccine safety are the primary drivers of hesitancy in HICs, in LMICs, fear of vaccine-associated harm is the most commonly reported reason for hesitancy [33]. As demonstrated by Figueroa et al. and Cuevas Barron et al., the ability of global health programs, such as the 100-Day Mission, to firmly establish inclusion, solidarity, and equity is essential to their legitimacy and effectiveness [7,32].

Additionally, several critical factors specific to LMICs must be carefully considered. These include regulatory aspects—such as pathogen sharing and oversight of new clinical trial sites—capacity building, and the need for social and political adaptation. Most importantly, sustained efforts are essential to educate not only healthcare professionals but the broader public.

Furthermore, within LMIC populations, significant internal disparities exist, where access to vaccines is uneven—often determined by factors such as economic status and geographic accessibility.

Lastly, in a landmark decision, the 78th World Health Assembly adopted the WHO Pandemic Agreement during its plenary session on 20 May 2025. The agreement was passed by a vote in the Committee, with 124 Member States in favor, none opposed, and 11 abstentions.

Key highlights include the following:The agreement’s adoption follows three years of intensive negotiations, initiated in response to the gaps and inequities revealed during the global COVID-19 response.It aims to strengthen global collaboration and ensure a more equitable, coordinated, and effective response to future pandemics.The following steps will focus on negotiations concerning the Pathogen Access and Benefit-Sharing (PABS) system.

The WHO Pandemic Agreement outlines key principles, strategies, and tools to improve international coordination for pandemic prevention, preparedness, and response. Central to this is ensuring equitable and timely access to vaccines, therapeutics, and diagnostics, particularly in times of global health emergencies [34].

## 4. Conclusions

The 100-Day Mission benefit includes a bold and necessary step forward regarding pandemic preparedness. Additionally, it is building off the lessons learned from the Spanish flu, AH1N1, and COVID-19 to shift how the world responds to new infectious diseases. The 100-Day Mission challenges in LMICs need to focus on speed, scalability, stakeholder and government commitment, and differ from HICs in order to achieve overall equity. The 100-Day Mission recommendations must be clearly conveyed in an increasingly interconnected world—yet one marked by stark global disparities—restructuring investments (through mechanisms such as Gavi, CEPI, the World Bank, and others) toward stronger surveillance, vaccine manufacturing, implementation, enhanced cooperation, improved communication, and greater political will, among other factors, will be essential to addressing the challenges of such an ambitious initiative in a more equitable and effective manner.

## Figures and Tables

**Figure 1 vaccines-13-00773-f001:**
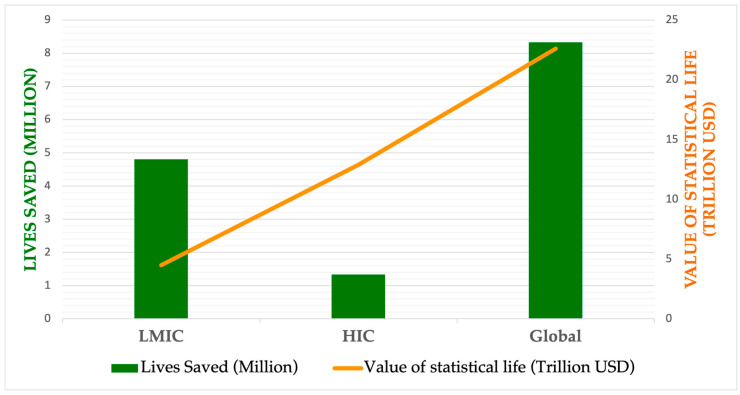
Estimated impact of lives saved and economic benefits across pandemic scenarios with prompt vaccine deployment (100-Day Mission). (Data obtained from Barnsley G, Olivera Mesa D, Hogan AB, Winskill P, Torkelson AA, Walker DG, et al. Impact of the 100-Day mission for vaccines on COVID-19: a mathematical modeling study. Lancet Glob Health. 2024 Nov;12(11): e1764–74.)

## Data Availability

The original contributions presented in this study are included in the article. Further inquiries can be directed to the corresponding author(s).

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
