# Peer review of "100-Day Mission for Future Pandemic Vaccines, Viewed Through the Lens of Low- and Middle-Income Countries (LMICs)"

_vaccines, 2025, doi:10.3390/vaccines13070773_

Round 1
Reviewer 1 Report
Comments and Suggestions for Authors
Abstract is well written and conveys the central idea of the review in a succinct manner.
- Line # 58, “This time, researchers developed a vaccine, but it proved insufficient to halt the early stages of the pandemic.” Could the author please provide a reference for this statement?
- Line # 59, “Millions became infected due to the delayed production and distribution of vaccinations, healthcare systems were strained, and economic losses escalated as infection and fear rates increased.” Could the author please provide a reference for this statement?
- Line # 74, “COVID-19 vaccines were authorized for emergency use less than a year after the virus's genetic sequence was studied, demonstrating the power of technology, innovation, and international cooperation”, Could the author please provide a reference for this statement?
- Could the authors please provide document link to reference # 17 (Coalition for Epidemic Preparedness Innovations Delivering Pandemic Vaccines in 100 Days 2022.)?
- Line #140-142, “To manage enormous amounts of genomic data, spot possible pandemic risks, and direct vaccine research efforts, sophisticated bioinformatics tools and data analytics are essential [10] “ is the reference # correct for this statement?
- Line # 150-152, “This invention boosts trust in vaccination deployment during crucial times while reducing delays [17].” Can the author please provide a link to this reference? Reviewer did a google search for “Coalition for Epidemic Preparedness Innovations Delivering Pandemic Vaccines in 100 Days 2022” and the document that was found does not support statements made in this paragraph.
- Line # 153-158, please provide link to the reference.
- 2.3 through 2.5 relies heavily on reference # 17. Could the authors be able to please provide other references.
- Line #181-188, unfortunately these lines have been copied from reference # 23 without any modifications.
- Could the author please explain how the data for Figure 1 was extracted from Barnsley G, Olivera Mesa D, Hogan AB, Winskill P, Torkelson AA, Walker 202 DG, et al. Impact of the 100-Day mission for vaccines on COVID-19: a mathematical modeling study. 203 Lancet Glob Health. 2024 Nov;12(11): e1764–74. Going through this reference the reviewer was unable to pinpoint this data.
- Line #221-236, unfortunately none of these points are explained through the lens of low- and middle-income countries.
- The reviewer is unable to find support to statement “Numerous studies have shown that success depends on political will, significant investments in public health infrastructure, and the guarantee of fair access to immunizations” in reference # 11, #18.
- Line #274-278, the reviewer would like to kindly point out that the issues discussed in these lines are not only associated with low- and middle-income countries.
Although the authors wish to steer the readers towards the problems associated with Low- and Middle-Income Countries, the theme of the article surrounds around data for United States and not much for low- and middle-income countries. It would be impactful if the authors truly layout the article write-up and the content from the point of view of lens of Low- and Middle-Income Countries. The challenges and impact of unpreparedness are not viewed through lens of Low- and Middle-Income Countries.
Author Response
We sincerely thank the reviewer for their thoughtful and constructive feedback.
All corresponding changes in the manuscript are highlighted in red text.
Comment 1: Line # 58, “This time, researchers developed a vaccine, but it proved insufficient to halt the early stages of the pandemic.” Could the author please provide a reference for this statement?
Response 1: It is reference number 3, sorry for the confusion. It’s been added.
Comment 2: Line # 59, “Millions became infected due to the delayed production and distribution of vaccinations, healthcare systems were strained, and economic losses escalated as infection and fear rates increased.” Could the author please provide a reference for this statement?
Response 2: It is reference number 3, sorry for the confusion. It’s been added.
Comment 3: Line # 74, “COVID-19 vaccines were authorized for emergency use less than a year after the virus's genetic sequence was studied, demonstrating the power of technology, innovation, and international cooperation”, Could the author please provide a reference for this statement?
Response 3: It is now reference number 7, sorry for the confusion. It’s been added.
Comment 4: Could the authors please provide document link to reference # 17 (Coalition for Epidemic Preparedness Innovations Delivering Pandemic Vaccines in 100 Days 2022.)?
Response 4: The link has been added; we apologize for not including it before, it is now reference 18.
Comment 5: Line #140-142, “To manage enormous amounts of genomic data, spot possible pandemic risks, and direct vaccine research efforts, sophisticated bioinformatics tools and data analytics are essential [10] “ is the reference # correct for this statement?
Response 5: Reference now number 11 is the right one for this sentence. This publication addresses this issue.
Comment 6: Line # 150-152, “This invention boosts trust in vaccination deployment during crucial times while reducing delays [17].” Can the author please provide a link to this reference? Reviewer did a google search for “Coalition for Epidemic Preparedness Innovations Delivering Pandemic Vaccines in 100 Days 2022” and the document that was found does not support statements made in this paragraph.
Response 6: The link has been added; we apologize for not including it before, it is now reference 18.
Comment 7: Line # 153-158, please provide link to the reference.
Response 7: The link has been added; we apologize for not including it before, it is now reference 18.
Comment 8: 2.3 through 2.5 relies heavily on reference # 17. Could the authors be able to please provide other references
Response 8: The link has been added; we apologize for not including it before, it is now reference 18.
Comment 9: Line #181-188, unfortunately these lines have been copied from reference # 23 without any modifications.
Response 9: We apologize, the text has been modified as suggested by the reviewer, that reference is now number 24.
Comment 10: Could the author please explain how the data for Figure 1 was extracted from Barnsley G, Olivera Mesa D, Hogan AB, Winskill P, Torkelson AA, Walker 202 DG, et al. Impact of the 100-Day mission for vaccines on COVID-19: a mathematical modeling study. 203 Lancet Glob Health. 2024 Nov;12(11): e1764–74. Going through this reference the reviewer was unable to pinpoint this data.
Response 10: By considering the reviewer’s comment on emphasizing the role of LMICs for our paper, we have changed completely figure #1, in which we explain the following:
“Figure 1 compares the estimated outcomes of vaccine deployment during the 100-Day Mission, highlighting both lives saved and financial gains in U.S. dollars. As illustrated, the number of averted deaths would be greater in low- and middle-income countries (LMICs) than in high-income countries (HICs). However, as expected—due to the higher value placed on statistical life in HICs—the projected financial gains from prompt vaccination are greater in those settings."
The data was obtained from two tables in the same reference.
We hope the reviewer is satisfied.
Comment 11: Line #221-236, unfortunately none of these points are explained through the lens of low- and middle-income countries.
Response 11: We appreciate the reviewer’s comment, and accordingly, have rephrased the initial and last sentence of that paragraph, emphasizing the need for more similar data in LMICs.
Comment 12: The reviewer is unable to find support to statement “Numerous studies have shown that success depends on political will, significant investments in public health infrastructure, and the guarantee of fair access to immunizations” in reference # 11, #18.
Response 12: Regarding reference number 18, which is now reference number 19, the publication includes multiple quotations that support our statement. Here are two illustrative examples: “Significant investments in public health infrastructure: Supporting the efforts of low- and middle-income countries to take full ownership of their national health security by developing the infrastructure and expertise to conduct epidemiological and clinical studies, support technology transfer, and establish national and regional vaccine manufacturing. Fair access to immunizations: CEPI's plan will not only accelerate vaccine development and reduce the threat posed by future epidemics and pandemics, it will also enable equitable access to these life-saving innovations for the most vulnerable populations. As a globally recognized organizing force for R&D collaboration and innovation, CEPI is uniquely placed to coordinate this international approach.”
Regarding reference 11, we have deleted that reference into that statement.
Comment 13: Line #274-278, the reviewer would like to kindly point out that the issues discussed in these lines are not only associated with low- and middle-income countries.
Response 13: We agree with the reviewer, we have changed that paragraph and also modified the following paragraph in order to make more emphasis on equity for LMICs.
Comment: Although the authors wish to steer the readers towards the problems associated with Low- and Middle-Income Countries, the theme of the article surrounds around data for United States and not much for low- and middle-income countries. It would be impactful if the authors truly layout the article write-up and the content from the point of view of lens of Low- and Middle-Income Countries. The challenges and impact of unpreparedness are not viewed through lens of Low- and Middle-Income Countries.
Response: We agree with the reviewer and have accordingly made substantial revisions to better reflect the LMICs perspective. This is particularly important, as this will be the first publication to emphasize the need for a 100-Day Mission from the viewpoint of low- and middle-income countries.
Reviewer 2 Report
Comments and Suggestions for Authors
Congratulations to the authors on this important manuscript that draws public attention to the 100-day mission as an important step forward in pandemic preparedness. The manuscript is well organized and highlights lessons from previous pandemics, primarily COVID-19, as well as the opportunities this strategy offers for an integrated response to future pandemics and health challenges but.
Author Response
We deeply appreciate the reviewer’s kind and enthusiastic words.
Reviewer 3 Report
Comments and Suggestions for Authors
Considering the increase in pandemics globally, this topic is very relevant to the general public, healthcare providers, policy makers, and program managers. The topic reflects the content of the manuscript. The abstract is an adequate summary of the manuscript. The historical background orients the readers to the topic. The aim of the manuscript is clearly stated. The vision of the 100-day mission is well discussed. The benefit across sectors, although clearly presented, should be restructured for readability.
MAJOR REVISIONS
- The benefits across sectors should be divided into benefits, challenges the mission is likely to face, and recommendations to address these challenges. Structuring the manuscript in this way will improve the relevance of the study and show what the authors are adding to the available literature.
MINOR REVISIONS.
- There is confusion on the use of the words vaccination, vaccines, and immunization.
- a) In line 60, the authors state, ‘Millions became infected due to the delayed production and distribution of vaccinations, …..’ ‘Vaccinations’ should be replaced with ‘vaccines.’
- b) In lines 63-64, the authors state, ‘Khazeni et al., estimate that if 20% of Americans had received vaccinations in October, …’ ‘Vaccinations’ should be replaced with ‘vaccines.’
- c) In lines 65-66, the authors state, ‘These figures demonstrate that even a slight increase in vaccination distribution can have a ….’ ‘Vaccination’ should be replaced with ‘vaccine.’
- d) In lines 76-77, the authors state, ‘Despite this, unequal vaccination distribution revealed political and economic divides, …..’ ‘Vaccination’ should be replaced with ‘vaccine.’
- e) In lines 88-89, the authors state, ‘….the delayed 88 vaccine rollout in 2009, and the uneven distribution of COVID-19 in 2021.’ Add ‘vaccines’ after COVID-19.
- f) In line 245, the authors state, ‘…..and the guarantee of fair access to immunizations [8,11,18,27].’ Replace ‘immunizations’ with ‘vaccines.’
- g) In line 261, the authors state, ‘Its main objective is to guarantee that vaccinations are prepared within 100 days….’ ‘Vaccinations’ should be replaced with ‘vaccines.’
- In lines 287-288, the authors state, ‘As demonstrated by Figueroa et al. and Cuevas Barron et al., the ability of global health programs….’ Add numerical references after the author names.
Author Response
We sincerely thank the reviewer for their thoughtful and constructive feedback.
All corresponding changes are in green text in the re-submission.
Response 1: Again, we appreciate the reviewer’s input, accordingly, we have completely rephrased our conclusion in the way the reviewer suggested.
Minor revisions
Comment 2: a) In line 60, the authors state, ‘Millions became infected due to the delayed production and distribution of vaccinations, …..’ ‘Vaccinations’ should be replaced with ‘vaccines.’
Response 2: It has been replaced.
Comment 3: b) In lines 63-64, the authors state, ‘Khazeni et al., estimate that if 20% of Americans had received vaccinations in October, …’ ‘Vaccinations’ should be replaced with ‘vaccines.’
Response 3: It has been replaced.
Comment 4: c) In lines 65-66, the authors state, ‘These figures demonstrate that even a slight increase in vaccination distribution can have a ….’ ‘Vaccination’ should be replaced with ‘vaccine.’
Response 4: It has been replaced.
Comment 5: d) In lines 76-77, the authors state, ‘Despite this, unequal vaccination distribution revealed political and economic divides, …..’ ‘Vaccination’ should be replaced with ‘vaccine.’
Response 5: It has been replaced.
Comment 6: e) In lines 88-89, the authors state, ‘….the delayed 88 vaccine rollout in 2009, and the uneven distribution of COVID-19 in 2021.’ Add ‘vaccines’ after COVID-19.
Response 6: It has been replaced.
Comment 7: f) In line 245, the authors state, ‘…..and the guarantee of fair access to immunizations [8,11,18,27].’ Replace ‘immunizations’ with ‘vaccines.’
Response 7: It has been replaced.
Comment 8: g) In line 261, the authors state, ‘Its main objective is to guarantee that vaccinations are prepared within 100 days….’ ‘Vaccinations’ should be replaced with ‘vaccines.’
Response 8: It has been replaced.
Comment 9:In lines 287-288, the authors state, ‘As demonstrated by Figueroa et al. and Cuevas Barron et al., the ability of global health programs….’ Add numerical references after the author names
Response 9: The numerical references have been added at the end of the corresponding sentence.
Reviewer 4 Report
Comments and Suggestions for Authors
As a part of CEPI’s 2022-2026 strategy, known as CEPI 2.0, to Prepare, Transform, and Connect the world, the 100-Day Mission was conceived to respond to the next Disease X threat by accelerating the development of safe, effective, and globally accessible vaccines in as little as 100 days. Much has already been published on the topic with particular reference to high-income countries (HICs). The authors of this paper stated that their goal was to highlight the mission’s implementation potential and challenges though the lens of low- and middle-income countries (LMICs). Though the paper is well-written and there are some discussions on the barriers LMICs face regarding equitable distribution of vaccines, most of the information presented in the paper apply to HICs that is repetitive from general literature. For example, in Section 2 on the vision of the 100-Day Misson, I do not find any pertinent description about how the LMICs are performing regarding “vaccine library” with pre-existing prototype vaccines, clinical trial infrastructure and readiness, identification of earlier biomarkers of immune response, and early characterization of pathogens. Regarding manufacturing capacity for vaccines, though majority of LMICs may have limited capacity, there are realistic examples of some countries that were not only producing vaccines for COVID-19 domestically but also helping in the international distribution of these vaccines. Within Benefits Across Sectors (Section 3), the authors did not produce specific data to document the economic impact of COVID-19 among LMICs across healthcare, education, tourism and hospitality sectors. Data for healthcare included an example from the U.S., and fair access to vaccines (lines 216-219 on page 5) do not even mention Gavi, the Vaccine Alliance organization, which aims to increase access to immunization in developing countries, and COVAX AMC, the financing instrument created by Gavi to support the participation of LMICs in the COVAX Facility. Furthermore, LMICs comprise a variety of countries across different continents that differ in their needs and financial abilities. For middle-income countries, the budgetary problems to effectively undertake the 100-Day Mission are expected to differ among lower- and upper-middle income countries. In this context, I saw a series of papers published in 2022 (End COVID-19 in low- and middle-income countries | Science) that could be useful for the focus of this review.
I am providing other minor comments below for the authors’ review.
There is no reference to SARS 2002-2003 and MERS in the Historical Background discussion. Though these were less severe in nature, they are often mentioned in any pandemic discussion. Also, SARS 2002-2003 in the U.S. mainly affected the children, and the economic loss estimates included the impact on parental productivity loss due to school closures. By contrast, COVID-19 mainly affected the elderly and the old, who had low immunity. These details could further enrich the discussion in this section.
In terms of comparative advantage of HICs in vaccine procurement, the authors could also include the strategy used by these countries to get special concessions from vaccine manufacturers through advance contracts and higher funding/subsidies that often led to stockpiling of COVID-19 vaccines in these countries inhibiting their distribution to LMICs.
The unequal access to vaccine coverage applies to both LMICs as a group and for populations within the LMICs. The limited money and other resources for basic health care coverage undermine the relative ability of LMICs to follow through the 100-Day Mission. The heterogeneities of LMCs in this and other ancillary areas could be highlighted by the authors.
There are different reasons for vaccine hesitancy in LMICs and HICs (please see: Nuwarda RF, Ramzan I, Weekes L, Kayser V. Vaccine Hesitancy: Contemporary Issues and Historical Background. Vaccines (Basel). 2022 Sep 22;10(10):1595. doi: 10.3390/vaccines10101595). The considerations of these additional details could show how the 100-Day Mission should be viewed through the lens of LMICs.
Author Response
We sincerely thank the reviewer for their thoughtful and constructive feedback.
All corresponding changes are in blue text in the re-submission.
Comment 1:
Response 1: The reviewer is absolutely correct in their observation. To date, vaccine libraries within the context of the 100-Day Mission remain largely absent in LMICs. We have expanded the corresponding paragraph to reflect this important point.
Comment 2:
Regarding manufacturing capacity for vaccines, though majority of LMICs may have limited capacity, there are realistic examples of some countries that were not only producing vaccines for COVID-19 domestically but also helping in the international distribution of these vaccines. Within Benefits Across Sectors (Section 3), the authors did not produce specific data to document the economic impact of COVID-19 among LMICs across healthcare, education, tourism and hospitality sectors. Data for healthcare included an example from the U.S., and fair access to vaccines (lines 216-219 on page 5) do not even mention Gavi, the Vaccine Alliance organization, which aims to increase access to immunization in developing countries, and COVAX AMC, the financing instrument created by Gavi to support the participation of LMICs in the COVAX Facility. Furthermore, LMICs comprise a variety of countries across different continents that differ in their needs and financial abilities. For middle-income countries, the budgetary problems to effectively undertake the 100-Day Mission are expected to differ among lower- and upper-middle income countries. In this context, I saw a series of papers published in 2022 (End COVID-19 in low- and middle-income countries | Science) that could be useful for the focus of this review.
Comment 2: We greatly appreciate the reviewer’s comments. However, our paper is not intended to focus on COVID-19 vaccination in LMICs. Rather, it aims to highlight the concerns from LMICs about the lack of readiness across multiple dimensions—particularly regarding the timely deployment of vaccines—in the event of a future pandemic caused by Pathogen X.
By considering the reviewer’s comment on emphasizing the role of LMICs for our paper, we have changed completely figure #1, in which we explain the following:
“Figure 1 compares the estimated outcomes of vaccine deployment during the 100-Day Mission, highlighting both lives saved and financial gains in U.S. dollars. As illustrated, the number of averted deaths would be greater in low- and middle-income countries (LMICs) than in high-income countries (HICs). However, as expected—due to the higher value placed on statistical life in HICs—the projected financial gains from prompt vaccination are greater in those settings.
Additionally, we have added Gavi, CEPI, the world Bank potential roles as investors in the last paragraph of our conclusions.
Comment 3: I am providing other minor comments below for the authors’ review.
There is no reference to SARS 2002-2003 and MERS in the Historical Background discussion. Though these were less severe in nature, they are often mentioned in any pandemic discussion. Also, SARS 2002-2003 in the U.S. mainly affected the children, and the economic loss estimates included the impact on parental productivity loss due to school closures. By contrast, COVID-19 mainly affected the elderly and the old, who had low immunity. These details could further enrich the discussion in this section.
Response 3: We deeply thank the reviewer’s input, we have added a whole paragraph on SARS 2002-03, with a new reference (number 6).
Comment 4: In terms of comparative advantage of HICs in vaccine procurement, the authors could also include the strategy used by these countries to get special concessions from vaccine manufacturers through advance contracts and higher funding/subsidies that often led to stockpiling of COVID-19 vaccines in these countries inhibiting their distribution to LMICs.
Response 4: Although the reviewer is correct in this statement, we cannot find a place to address this in the context of the 100-day mission, since we have already mentioned the disparities between LMICs and HICs. We apologize for not being able to address this in particular.
Comment 5: The unequal access to vaccine coverage applies to both LMICs as a group and for populations within the LMICs. The limited money and other resources for basic health care coverage undermine the relative ability of LMICs to follow through the 100-Day Mission. The heterogeneities of LMCs in this and other ancillary areas could be highlighted by the authors.
Response 5: The reviewer is absolutely right, and we have added the following in our discussion:
“Furthermore, within LMIC populations, significant internal disparities exist, where access to vaccines is uneven—often determined by factors such as economic status and geographic accessibility.”
Comment 6: There are different reasons for vaccine hesitancy in LMICs and HICs (please see: Nuwarda RF, Ramzan I, Weekes L, Kayser V. Vaccine Hesitancy: Contemporary Issues and Historical Background. Vaccines (Basel). 2022 Sep 22;10(10):1595. doi: 10.3390/vaccines10101595). The considerations of these additional details could show how the 100-Day Mission should be viewed through the lens of LMICs.
Response 6: The reviewer’s suggestion is of great relevance, and we have added the following with an added reference (number 32):
“While concerns about vaccine safety are the primary drivers of hesitancy in high-income countries (HICs), in low- and middle-income countries (LMICs), fear of vaccine-associated harm is the most commonly reported reason for hesitancy.”
Round 2
Reviewer 1 Report
Comments and Suggestions for Authors
Thank you for making updates in the manuscript. However, the reviewer did not find significant improvement in the manuscript.
As mentioned earlier,
The challenges and impact of unpreparedness are not viewed through lens of Low- and Middle-Income Countries. The article, while reading, gives a nice view about the 100-day mission plan and its importance. However, only few sentences about LMIC have been incorporated. Instead, the author should have added a paragraph after each section with a perspective of LMIC towards that section.
Thanks!
Author Response
We sincerely thank the reviewer for their thoughtful and constructive feedback.
All corresponding changes in the manuscript are highlighted in red text.
Comment 1: The challenges and impact of unpreparedness are not viewed through lens of Low- and Middle-Income Countries. The article, while reading, gives a nice view about the 100-day mission plan and its importance. However, only few sentences about LMIC have been incorporated. Instead, the author should have added a paragraph after each section with a perspective of LMIC towards that section.
Response 1: We sincerely appreciate the reviewer’s interest in strengthening our manuscript. In response, we have added several sentences to better emphasize the 100-Day Plan from the perspective of LMICs, as suggested. Additionally, in the conclusion section (not marked in red), we place particular emphasis on the perspectives of low- and middle-income countries (LMICs). We hope these revisions meet the reviewer’s expectations.
Reviewer 3 Report
Comments and Suggestions for Authors
Thank you for addressing my comments. However, the conclusion should flow. There should be complete sentences, e.g. 'The 100 day Mission challenges include ....'
Author Response
We sincerely thank the reviewer for their thoughtful and constructive feedback.
All corresponding changes are in green text in the re-submission.
Comment 1: Thank you for addressing my comments. However, the conclusion should flow. There should be complete sentences, e.g. 'The 100 day Mission challenges include ....'
Response 1: We greatly appreciate the reviewer’s thoughtful comments and suggestions. In response, we have revised the conclusions to improve their clarity and flow, as recommended.
Reviewer 4 Report
Comments and Suggestions for Authors
On page 6, line 255, please change LMIC to LMICs.
In the Conclusion section, I am not aware about the journal policy regarding broad sections. If there is any need, this could be fixed by minor changes.
Author Response
We sincerely thank the reviewer for their thoughtful and constructive feedback.
All corresponding changes are in blue text in the re-submission.
Comment 1: On page 6, line 255, please change LMIC to LMICs.
Response 1: We do apologize; the change has been made.
Comment 2: In the Conclusion section, I am not aware about the journal policy regarding broad sections. If there is any need, this could be fixed by minor changes.
Response 2: We thank the reviewer for highlighting this point. Accordingly, we have revised the conclusions to enhance their clarity and flow.
Round 3
Reviewer 1 Report
Comments and Suggestions for Authors
Thank you for incorporating the suggestions.